# Using a Quantitative High-Throughput Screening Platform to Identify Molecular Targets and Compounds as Repurposing Candidates for Endometriosis

**DOI:** 10.3390/biom13060965

**Published:** 2023-06-08

**Authors:** Molly L. Churchill, Sarah J. Holdsworth-Carson, Karla J. Cowley, Jennii Luu, Kaylene J. Simpson, Martin Healey, Peter A. W. Rogers, J. F. Donoghue

**Affiliations:** 1Gynaecology Research Centre, Department of Obstetrics and Gynaecology, University of Melbourne and The Royal Women’s Hospital, Parkville, VIC 3052, Australia; molly.churchill@outlook.com (M.L.C.); sarah.carson@epworth.org.au (S.J.H.-C.); parogers@unimelb.edu.au (P.A.W.R.); 2Julia Argyrou Endometriosis Centre, Epworth HealthCare, Richmond, VIC 3121, Australia; 3Victorian Centre for Functional Genomics, Peter MacCallum Cancer Centre, Parkville, VIC 3010, Australiakaylene.simpson@petermac.org (K.J.S.); 4Sir Peter MacCallum Department of Oncology, University of Melbourne, Parkville, VIC 3010, Australia; 5Gynaecology Endometriosis and Pelvic Pain Unit, Royal Women’s Hospital, Parkville, VIC 3052, Australia

**Keywords:** endometriosis, endometrial stromal cells, estrogen-signalling pathways, high-throughput screening, high-content imaging

## Abstract

Endometriosis, defined as the growth of hormonally responsive endometrial-like tissue outside of the uterine cavity, is an estrogen-dependent, chronic, pro-inflammatory disease that affects up to 11.4% of women of reproductive age and gender-diverse people with a uterus. At present, there is no long-term cure, and the identification of new therapies that provide a high level of efficacy and favourable long-term safety profiles with rapid clinical access are a priority. In this study, quantitative high-throughput compound screens of 3517 clinically approved compounds were performed on patient-derived immortalized human endometrial stromal cell lines. Following assay optimization and compound criteria selection, a high-throughput screening protocol was developed to enable the identification of compounds that interfered with estrogen-stimulated cell growth. From these screens, 23 novel compounds were identified, in addition to their molecular targets and in silico cell-signalling pathways, which included the neuroactive ligand–receptor interaction pathway, metabolic pathways, and cancer-associated pathways. This study demonstrates for the first time the feasibility of performing large compound screens for the identification of new translatable therapeutics and the improved characterization of endometriosis molecular pathophysiology. Further investigation of the molecular targets identified herein will help uncover new mechanisms involved in the establishment, symptomology, and progression of endometriosis.

## 1. Introduction

Endometriosis is an estrogen-dependent gynaecological disease that affects up to 11.4% women of reproductive age [1,2] as well as trans and gender-diverse people (for whom statistics are currently unavailable). In extremely rare conditions such as Rokitansky–Küster–Hauser (MRKH) syndrome (congenital aplasia of the vagina, cervix, and uterus), endometriomas have been identified, most likely due to congenital remnants of endometrial tissue [3]. Further to this, endometriosis of the bladder, peritoneum, and genitals has been described in men [4]. While more substantial evidence of endometriotic lesions in males is needed, these rare cases may be demonstrations of remnant embryological cell growth [4] or mesothelium metaplasia [5] in response to elevated estrogen. Endometriosis is characterized by lesions comprising endometrial-like cells (stromal and epithelial), which grow most commonly within the peritoneal cavity, stimulated by estrogen [6,7]. Symptoms of endometriosis commonly include dysmenorrhea, chronic pelvic pain, subfertility, and reduced quality of life. At present, there is no long-term cure, with an average 5-year recurrence rate following surgery of 20.5 to 43.5% [8,9,10]. Current treatments involve surgery to remove lesions or medical approaches including non-steroidal anti-inflammatories, analgesics, hormonal contraceptives, gonadotropin-releasing hormone (GnRH) agonists, and others, such as synthetic androgens to reduce systemic estrogen levels [11]. Patient responsiveness to these treatments is variable, with serious and undesirable consequences such as hypestrogenism, early menopause, and infertility [11]. It is therefore a priority that new therapeutic options are identified that do not directly inhibit estrogen or the estrogen receptor (ER).

Endometriosis is dependent upon estrogen signalling, which controls several physiological processes by directly and indirectly regulating gene transcription via the genomic and non-genomic estrogen signalling pathways. The genomic pathway is mediated by two estrogen receptors (ERs), ERα and Erβ, which either directly translocate to the nucleus and bind estrogen response elements (EREs) in or near the promoters of target genes or indirectly via transcription factors that interact with gene promoters [12,13]. A third member of the ER family is membrane-bound G-protein-coupled ER1 (GPER1), formally known as GPR30. This receptor, in addition to membrane-bound ERα and ERβ, induces non-genomic estrogen signalling pathways that rapidly respond to estrogen to generate signal transduction cascades that activate PI3K/AKT, MAPK, cAMP, and others [14,15,16]. Further, estrogen also signals through GPR30 independent of ER and transactivates tyrosine kinase receptors such as EGFR to indirectly regulate gene expression through the activation of cell-signalling factors and transcription factors [17]. Intracellular signalling cascades also mobilise Ca^2+^, cAMP production, NO production, and cytoskeletal rearrangements [13,18]. Taken together, these intracellular signalling cascade elements downstream of estrogen and its receptors are therefore ideal candidates for the indirect targeting of estrogen signalling pathways via therapeutic compound screens.

The high-throughput screening (HTS) of compounds is an increasingly popular approach to in vitro drug discovery as it reduces the time and cost of drug development compared to conventional target-based discovery [19,20]. HTS involves the rapid investigation of thousands of compounds with the use of 384- or 1536-well plates and the automation of cell and compound dispensing [19]. HTS also provides an opportunity to repurpose pre-existing compounds due to previously established clinical safety parameters, which substantially fast-tracks clinical trials, drug approval, and patient access [21,22,23,24]. Therefore, applying HTS to thousands of compounds in eutopic endometrial stromal cell lines may result in the discovery of compounds that can be repurposed for a condition lacking an efficacious drug choice. The HTS may also provide new mechanistic insights into the pathogenesis of endometriosis and support the development of new diagnostics and therapeutics. As endometriosis is an estrogen-dependent disease, the inhibition of cell proliferation in an HTS compound screen must be performed under estrogenic conditions to best mimic the endometriosis environment and to ensure the compounds identified are targeting the downstream effectors of estrogen stimulation and not estrogen or its receptor directly. Therefore, the aim of this study was to develop an HTS protocol using endometriosis-patient-derived immortalized endometrial stromal cell lines for the identification of non-estrogen-receptor-targeting compounds that have translational potential as therapeutics for endometriosis.

## 2. Materials and Methods

### 2.1. Reagents and Kits

Phenol-red-free DMEMF/12 (PRF-DMEM/F12), charcoal-stripped FBS (CS-FBS), penicillin–streptomycin, GlutaMAX-I, insulin-transferrin-selenium (ITS), Dulbecco’s phosphate-buffered saline (DPBS), TrypLE express, paraformaldehyde (PFA), 4′,6-Diamidino-2-Phenylindole, Dihydrochloride (DAPI), and Cell Tracker™ Green 5-chloromethylfluorescein diacetate (CMFDA) were purchased from Thermo Fisher Scientific, Waltham, MA, USA. Dimethyl sulphoxide (DMSO), Triton X-100, and estradiol-17β (estrogen) were purchased from Sigma-Aldrich, Castle Hill NSW 2154, Australia. The Real Time-Glo™ MT Cell Viability Assay was purchased from Promega, Alexandria NSW 2015, Australia. Salinomycin, panobinostat, fulvestrant, tamoxifen, and Y26732 were purchased from Selleck Chem, Houston, TX 77014 USA.

### 2.2. Cell Lines and Culture

Two hTERT immortalized endometriosis-patient-derived hESC cell lines were screened [25]. These cell lines were generated from endometrial tissue collected via curettage from women who had provided informed consent prior to undergoing laparoscopy at the Royal Women’s Hospital, Melbourne. The ethical approval was provided by the Human Research Ethics Committee (HREC 10–43 and HREC 11–24) at the Royal Women’s Hospital, Melbourne, Australia. All experiments were performed in accordance with relevant guidelines and regulations. The 1455 hESC line was derived from a stage II endometriosis patient with no pain symptomology and no reported recurrent disease, while the 1458 hESC line was derived from a stage IV endometriosis patient with severe pain and recurrent disease. The cell lines were confirmed to be free of mycoplasma prior to their use and were cultured in PRF-DMEM/F12/10% CS-FBS/1 × penicillin–streptomycin/2 mM GlutaMAX-I/1× ITS at 37 °C in a 5% CO_2_ humidified incubator. For high HTS, large stocks of cells were grown (P10-P20) and frozen in batches of 20 vials of 1 × 10^6^ cells per vial. This was carried out to ensure that each compound plate screened was performed on the same passage of cells over several weeks.

### 2.3. Compound Libraries

Assay-ready, pre-stamped 384-well compound library plates were sourced from an open access compound library from Compounds Australia (Griffith University, Nathan QLD 4111, Australia). These plates were from the CA-FDA Collection, the CA-Epigenetics Collection, and the CA-Kinase Collection. Each plate left columns 23 and 24 reserved empty for the addition of in-house quality control (QC) compounds. The compound library contained 3517 compounds selected from the FDA clinically approved drug and compound library and the Selleck Kinase and Epigenetics drug libraries. Each pre-stamped compound plate contained 50 µM (5× final concentration) of compound in a 650 nL aliquot. On the day of their use, pre-stamped compound plates were reconstituted with 65 µL/well of PRF-DMEM/F12 + 5% CS-FBS using a BioTek EL406 dispenser (5 µL cassette, high flow rate) (Biotek™ Winooski, VT, 05404-1356, USA) (Appendix A).

### 2.4. High-Throughput Screening (HTS)

Cells (800 cells/well) were seeded in 384-well black-walled clear-base plates (40 µL/well) (Corning Incorporated, Corning, NY 14831, USA) in PRF-DMEM/F12 + 10% CS-FBS using a Biotek™ EL406 dispenser. The seeded plates were pulse-centrifuged briefly and incubated for 24 h in a LiCONiC high-throughput incubator (LiCONiC Instruments, Liechtenstein). Following cell adhesion, spent media were removed using a BioTek™ wash-aspirator manifold and replaced with fresh media (40 µL) supplemented with 5% CS-FBS with 0.3% DMSO (vehicle) or 10^−5^ M estradiol-17β (estrogen) (5 µL cassette, high flow rate; aspirator coordinates: Z = 36). The compounds were then added as described below.

### 2.5. Quality Control (QC) Compound Screen

QC positive (QC^pos^) control compounds *inhibit* cell survival (the outcome desired for the HTS), while the QC negative (QC^neg^) control compounds *promote* cell survival (the outcome not desired for the HTS). In plates pre-seeded with the 1455 or 1458 cells, the initial QC compounds, panobinostat (HDAC inhibitor), salinomycin, (antibiotic), fulvestrant, tamoxifen (selective estrogen receptor modulators), and Y26732 (ROCK inhibitor) were serially diluted (1:2 for 10 dose points from 50 µM to 0.05 µM) in triplicate. The cells were cultured for 3 days under both vehicle and estrogen conditions using the HTS protocol as described above (panobinostat, salinomycin, and Y26732), while the Real Time-Glo™ MT Cell Viability Assay was used for fulvestrant and tamoxifen according to the manufacturer’s instructions (Promega). Acceptable QC^neg^ control compounds exhibited no inhibition of cell growth during estrogen treatment, while the QC^pos^ controls exhibited >70% inhibition of cell growth at 10 µM during estrogen treatment [26,27].

### 2.6. Compound Library Screen

Pre-stamped compound plates were reconstituted (described above), and the QC compounds (50 µM) were added manually (65 µL/well) into columns 23 and 24 in a randomized layout including 8 wells of DMSO (1.5%), 4 wells of media supplemented with 5% CS-FBS, and 4 wells of each QC compound. All compounds were then dispensed (10 µL/well) using the ALH 3000 Advanced Liquid Handling system (Caliper Life Sciences, Hopkinton, MA 01748, USA). The final compound concentrations were 10 µM in DMSO (0.3%) in a total volume of 50 µL.

Cell plates and compounds were incubated for 3 additional days, after which the media were removed, and the cells were washed with DPBS. Using the BioTek EL406 plate wash/dispense automation, the cells were then live-cell stained with 4 µM of green CMFDA in serum-free media (25 µL/well, 30 min, 37 °C), fixed with 2% PFA in DPBS (25 µL/well, 15 min, room temperature [RT]), washed with DPBS (50 µL/well), and then stained with DAPI nuclear stain (0.005 mg/mL in 50 mM Tris/PBS + 10% Triton X-100; 25 µL/well, 20 min, RT). The cells were then washed twice and mounted in DPBS. Each plate was thermo-sealed with foil (Plateloc, Agilent, CA, USA) and was thus ready for imaging.

### 2.7. High-Content Imaging (HCI)

The wells were imaged with cell membranes and nuclei detected via segmentation; the cell nuclei were quantified using a fully automated Cellomics ArrayScan™ VTi High Content Imager (Thermo Fisher Scientific). An image acquisition software (Cellomics Scan software, v7.6.2.1, Thermo Fisher Scientific) was used to capture 16 fields per well using a 10× objective to ensure >90% of cells were counted (Appendix A). The “Target Activation V4” bioapplication (Cellomics, Thermo Fisher Scientific) was used with thresholding to identify valid nuclei, which included all objects with a total area between 60 and 1400 pixels and an average intensity of less than 4095. The images were manually checked for acquisition errors. This included assessing the image sharpness, nuclei tracing, low DMSO control cell counts, and artefacts. If any of these errors were detected, the wells were removed from the analysis.

### 2.8. Selection of Compound Hits

Cell counts were normalized to the mean cell counts of the vehicle control wells on a per-plate basis. Outlier control wells were removed from analysis if they were 3 standard deviations (SD) above or below the mean of all vehicle control wells per plate. If the cell viability in the vehicle test wells fell below 50% of the vehicle-alone wells (an indication of compound toxicity in the absence of estradiol-17β), the wells were removed from all further analysis. The percentage of estrogen-driven cell growth inhibition (%E-DI) was determined by the following equation: %E-DI = 100 (β/α), where β is the normalized cell count of estrogen-treated cells, and α is the normalized cell count of vehicle-treated cells (Appendix A). The %E-DI cut-off for compound hit determination was calculated based on the average %E-DI + 2.5 standard deviations (SDs) of all samples tested [28].

The compound “hits” that were identified in the initial HTS for confirmation were individually (cherry) picked and stamped onto 384-well compound plates by Compounds Australia, leaving columns 23 and 24 blank for QC compound additions as previously described. Each compound was serially diluted (1:5 for 5 dose points from 10 µM to 0.02 µM) to determine the IC_50_ (Appendix A). The plates were reconstituted in 65 µL of media and dispensed onto 1455 and 1458 pre-seeded cell plates and cultured for 3 days, followed by staining, fixation, imaging, and quantification as previously described. To determine the IC_50_ of each compound, the % survival was plotted against the log compound concentration (log µM) using a nonlinear regression in GraphPad Prism 7. Compounds with an IC_50_ ≤ 1 µM were designated lead compounds.

### 2.9. In Silico Analysis of Compound Hits

Lists of molecular targets for each compound hit were assembled using public bioinformatics databases: Drug Bank [29] and SwissTargetPrediction [30]. Protein–protein interaction (PPI), Gene Ontology (GO), and Kyoto Encyclopedia of Genes and Genomes (KEGG) pathway analyses were performed using the online String Pathway Database [31].

### 2.10. Data and Statistical Analysis

After normalization, the assay quality and robustness were determined using the Z′ factor and %CV. To determine the Z′ factor for the antagonist/inhibition assay, the following equation was used: Z′ factor = 1 − 3(*δn* + *δp*)/(*µn* − *µp*), where *δ* = standard deviation, *µ* = mean, *n* = negative control (MAX value), and *p* = positive control (MIN value), respectively. A Z′ factor = 1 indicates an ideal assay in which the SD dynamic range is extremely large. A Z′ score between 0.5 and 1 indicates an excellent assay as the SD dynamic range was large enough to consider test compounds as real hits. A Z′ factor between 0 and 0.5 indicates a marginal but acceptable assay, and a Z′ < 0 indicates an unacceptable assay [26]. This metric was determined for all control wells during the analysis. The %CV is a measurement of sample variability that measures the ratio between the standard deviation and the mean of a sample. The lower the value, the less variation occurs, demonstrating reliable reproducibility. To determine the %CV (coefficient of variance) [26,27], the following equation was used: %CV = 100 (*δ*/*µ*), where *δ* is the standard deviation, and *µ* is the mean of the sample. An arbitrary threshold was set at %CV < 25 for an HTS to be considered acceptable, while a %CV > 25 was considered unacceptable.

## 3. Results

### 3.1. Establishing the HTS Quality Control Parameters for Estrogen-Stimulated Cell Survival

Two immortalized hESC cell lines were used for the HTS compound screen. The 1455 and 1458 hESC lines were previously described and are known to be hormonally responsive [25]. For this screen, the hESC cells were plated out into 384-well clear-bottom plates (800 cells/well) and after 24 h, DMSO, estradiol-17β, and a series of quality control (QC) compounds were added in replicate. The QC control compounds were selected based upon a strong reliable inhibition or promotion of cell survival over several wells and plates under estrogen and vehicle conditions. QC control compounds that caused estrogen-specific responses were excluded. This requirement was essential to ensure that the HTS could be reliably performed over several weeks due to the size of the library used (3517 compounds) and the different conditions required for the screen. At a single dose of 10 µM, panobinostat and salinomycin were identified as positive QC (QC^pos^) control compounds as they *inhibited* cell survival—the outcome required for HTS compound hits (Appendix A–D). Panobinostat is a pan-HDAC inhibitor that inhibits epigenetic activity, while salinomycin is an antibiotic that interferes with membrane ion concentrations, resulting in osmotic pressure imbalances and cell death [32]. Both compounds reliably inhibited cell survival in a dose-dependent manner across multiple wells and plates, including the HTS dose of 10 µM. 

At 10 µM, fulvestrant was identified as the only reliable negative QC (QC^neg^) control compound compared to tamoxifen and Y26732, as it reliably promoted cell survival—the unwanted outcome for HTS compound hits (Appendix A–J). While fulvestrant is an anti-estrogenic compound that reduces cell survival by degrading the estrogen receptor, at 10 µM, cell survival was not compromised for these two cell lines. Tamoxifen, another anti-estrogenic compound selectively inhibited cell survival during estrogen treatment compared to the vehicle and was therefore excluded. While Y26732 is a ROCK inhibitor that promotes cell survival across different concentrations, it was not reliably mitogenic across several wells and plates for these cell lines. For these reasons, only panobinostat, salinomycin, and fulvestrant were used as reliable QC compounds for the initial HTS (Figure 1).

### 3.2. Demonstrating HTS Assay Robustness

Following the normalization of the QC wells, the Z-prime (Z’) factor and percentage coefficient of variance (%CV) were determined. The Z′ factors for all plates, conditions, and cell lines were ≥0.5 (Appendix A), demonstrating good separation between QC^pos^ and QC^neg^ [26]. The only exceptions to this were the 1455 tamoxifen wells, which fell below 0.3, indicating that tamoxifen was not suitable as a QC control compound for HTS, confirming previous findings. The %CVs of the QC^neg^ control compounds (DMSO, fulvestrant, and Y26732) for all plates, conditions, and cell lines were <12% (Appendix A), well below our set threshold of <25% and the commonly published thresholds of 10–20% [33,34]. The %CVs of the QC^pos^ control compounds (Panobinostat and salinomycin) for all plates, conditions, and cell lines were <22%, also below our set threshold of <25%. Again, tamoxifen proved to be an unsuitable QC control compound as the %CV ranged from 19.29 to 27.74%. These data established the initial HTS QC^neg^ and QC^pos^ compounds and a reliably robust HTS protocol.

### 3.3. Identification of Compounds That Specifically Inhibited Estrogen-Stimulated Cell Survival

Samples with more than 50% reduced cell viability in vehicle wells were excluded from further analysis. This initial triage was used to ensure that only compounds that were inhibitory during estrogen treatment were considered for analysis. Following this, 283 compounds (8.1%) were removed from analysis, leaving 3234 compounds (91.9%) for analysis. To further triage this large list of compounds, a threshold of 70% inhibition of cell survival was set. This threshold was derived from the averaged percentage of estrogen-dependent inhibition (%E-DI) + 2.5 standard deviations. At this threshold, 19 compounds were identified for the 1455 cell line (Figure 2A), and 36 compounds were identified for the 1458 cell line (Figure 2B). Together, the HTS identified 55 compounds (1.7%) of interest for further validation.

### 3.4. Identification of Lead Compounds Using Quantitative HTS (qHTS)

A quantitative HTS (qHTS) was used to serially dilute the 55 compounds identified in a 5-point dose response to generate a half-maximal inhibitory concentration (IC_50_). The first screen was without replicate samples, while the second screen was performed with duplicates. An average of the two independent screens was then used for all calculations. To ensure assay confidence, two extra QC^neg^ control compounds were included in this screen. These two compounds were identified from the initial HTS as they maintained cell viability during the vehicle and estradiol-17β treatments more consistently than Y26732. These compounds were SNO1004380 (beta-adrenergic agonist) and SNO1006318 (steroid sapogenin) (Figure 2C,D). Therefore, the qHTS included the following control wells: 0.03% DMSO (compound solute), SNO1004380, SNO1006318, and fulvestrant as QC^neg^ control compounds and salinomycin as the QC^pos^ control compound. As salinomycin was consistently inhibitory to cell growth, panobinostat was no longer required. The plating efficiency and cell counts for each replicate were good with small SEM, and good separation between positive and negative QC control compounds was observed (Figure 2E).

The Z′ factors for all plates, conditions, and cell lines were ≥0.5 except for the 1455 fulvestrant vehicle wells (0.43) (Appendix A). The %CV values of the QC^neg^ control compounds (DMSO, fulvestrant, SNO1004380, and SNO1006318) for all plates, conditions, and cell lines were <13% (threshold set to <25%) (Appendix A). The two new QC^neg^ control compounds had Z′ factors and %CVs consistent with the DMSO data from the initial HTS, making these two new QC compounds suitable for future HTS. The %CV values of the QC^pos^ control compound (salinomycin) for all plates, conditions, and cell lines were <20%, also below our set threshold of <25%. Together, these data indicate the qHTS was performing within the previously established assay parameters, enabling the identification of lead compounds from this qHTS.

From the first qHTS (samples without replicates), lead compounds were triaged using the calculated IC_50_, which was derived from the 5-point dilution curves. An IC_50_ (<1 µM) was employed as an arbitrary cut-off to reduce the identification of non-specific toxic compounds that would have no clinical or translational relevance. Twenty-three (23) compounds were identified from the initial 5-point dilution curves and used in the second qHTS (samples in duplicate) to confirm IC_50_ data. From this screen, four compounds were identified as hits for the 1455 hESC cell line only (Aminothiazole, Hydroxyzine Pamoate, Ketotifen Fumarate, and Spectinomycin HCl), six compounds were identified as hits for the 1458 hESC cell line only (Benazepril HCl, Ceftibuten, Chloroquinalol, Editol, Repaglinide, and Silenafil Citrate), and seven compounds were identified as hits for both cell lines (7,8-Dimethoxyflavone, Chlordiazepoxide, Cytidine triphosphate disodium, Indoprofen, Pantoprazole, Pregabalin, and Promazine HCl) (Table 1). Six compounds were excluded due to a non-determination (ND) of the IC_50_ (or IC_50_ > 1 µM) (Ethynodiol diacetate, Baicalin, Penicillamine, Fomepizole, Ergotamine, and Iodoquinol).

### 3.5. In Silico Analysis of Compound Molecular Targets and Pathways

The final 17 lead compounds identified were entered into the online drug and compound databases (see methods section), to determine which molecular targets (proteins/receptors, etc.) were known for each compound. Lists of the molecular targets were uploaded into the STRING protein pathway analysis database to determine protein interactions and the biological pathways utilized the by cell lines.

From the four compounds identified as hits for the 1455 hESC line, nine molecular targets were identified. To perform the string analysis, extra intuitive interactive targets were generated by the STRING program (Figure 3; Appendix A). The most significant functional enrichment pathways were the GO:0050999 regulation of nitric oxide synthase activity (FDR 4.44 × 10^−10^), the GO:0009051 pentose-phosphate shunt, oxidative branch (FDR 1.42 × 10^−07^), the GO:0055114 oxidation reduction process (FDR 7.75 × 10^−06^), and HSA1430728 metabolism (FDR 2.40 × 10^−05^). The most significant KEGG signalling pathways were prolactin signalling (FDR 8.54 × 10^−06^), estrogen signalling (FDR 0.00069), and VEGF signalling (FDR0.0413).

From the six compounds identified as hits for the 1458 line only, 92 molecular targets were identified (Figure 4; Appendix A). The most significant functional enrichment pathways were HSA143078 metabolism (FDR 5.39 × 10^−09^), HSA162582 signal transduction (FDR 1.42 × 10^−06^), HSA168249 innate immune system (FDR 0.0061), and apoptosis (FDR 0.0427). The most significant KEGG signalling pathways were the cAMP signalling (FDR 3.73 × 10^−05^), PI3K-Akt signalling (FDR 0.00083), calcium signalling (FDR 0.00081), p53 signalling (FDR 0.00067), cGMP-PKG signalling (FDR 0.00039), HIF-1 signalling (FDR 0.00039), insulin signalling (FDR 0.0046), estrogen signalling (FDR 0.0046), TNF signalling (FDR 0.0131), FoxO signalling (FDR 0.0186), PPAR signalling (FDR 0.0278), and IL-17 signalling (0.0435) pathways.

From the seven compounds identified as hits for both stromal cell lines, 40 molecular targets were identified (Figure 5; Appendix A). The most significant functional enrichment pathways were the hsa04080 neuroactive ligand–receptor interactions (FDR 1.19 × 10^−16^), hsa00591 linoleic acid metabolism (FDR 2.84 × 10^−05^), hsa01100 metabolic pathways (FDR 0.0220), and hsa04010 MAPK signalling (FDR 0.0342) pathways. The most significant KEGG signalling pathways were the calcium signalling pathway (FDR 1.34 × 10^−11^), cAMP signalling pathway (FDR 0.00013), MAPK signalling pathway (FDR 0.0342), and cGMP-PKG signalling pathway (FDR 0.0496).

## 4. Discussion

This study presents the first high-throughput screen of large compound libraries using patient-derived endometrial stromal cells to identify new clinically translatable therapeutics for endometriosis. While the compounds identified from this HTS show promise, further evaluation will be required before clinical translation to demonstrate their efficacy and specificity as anti-endometriosis agents. HTS of large compound libraries is proving to be a valuable tool for drug discovery and drug repurposing in a variety of diseases [19]. For endometriosis, the development of HTS technologies has been limited to small-molecule drug/compound screens using the in silico analysis of predicted endometriosis targets [35,36,37] and high-throughput RNA sequencing [38,39] to identify new endometriosis “druggable targets”. The use of patient-derived endometrial stromal cell lines in this HTS was made possible through protocol miniaturization and the identification of robust QC compounds. This protocol also benefited from the adaptable optimization of each screen [40]. This process allowed for the identification of new negative QC compounds for inclusion in the final confirmation HTS to improve assay robustness. Using adaptable optimization, the Z′ factor and the %CV were improved across all assay plates and reduced the risk of false discovery.

The qHTS was performed under estrogen stimulation to mimic the diseased state of endometriosis. In doing so, changes in gene expression induced by estrogen [41] result in compound targets that may be selectively expressed in ectopic lesions where estrogen signalling is aberrantly upregulated [42]. From this study, 17 compounds were identified that targeted a diverse range of cell surface receptors, cell-signalling molecules, and metabolic pathways, and the compound classifications ranged from antibiotic, anti-depressant, anti-psychotic, anti-histamine, anti-diabetic, and anti-convulsive, to other compounds. Several of the compounds identified targeted molecules involved in endometriosis-associated processes, including downstream estrogen-signalling molecules (PI3K/AKT, NO, cAMP) (7–8 Dimethoxyflavone, and Sildenafil citrate), inflammation (Aminothiazole, Cytidine triphosphate disodium, 7–8 Dimethoxyflavone, Ketotifen Fumarate, and Sildenafil Citrate), and neurotransmitter pathways (Hydroxyzine Pamoate, Pregabalin, Promazine, and Chlordiazepoxide). Together, these findings suggest that the endometrial stromal cells may contribute to the inflammation, neuropathic pain, and nociception associated with endometriosis through the expression of pro-inflammatory cytokines [43] and neuroreceptors [44]. At present, there are no reports of these compounds being used to treat endometriosis.

From the in-silico data, the most significant KEGG pathway identified was the neuroactive ligand–receptor interaction that was identified from compounds that suppressed the growth of both cell lines during estrogen treatment. These neuroactive receptors included histamine receptors (HRH1), serotonin receptors (HTR_2A_) and dopamine receptors (DRD1–3). The expression of neuroactive receptors on non-neuronal cells is not new [44,45]. Histamine receptors have previously been reported on myometrial smooth muscle cells [46], endothelial cells [47], and reproductive tissues [48]. While histamine (HRH ligand) has been shown to stimulate estrogen production in granulosa cells [49], estrogen also increases histamine secretion and the release of proinflammatory cytokines from mast cells [50,51,52]. Further, the placental mast cell release of histamines promotes myometrial contraction during labour via the HRH1 receptors [53]. Together, the histamine receptor (and histamines) may have roles in proinflammatory responses and vascular tone in endometriosis.

Serotonin and its receptor (HTR) are well-described neurotransmitters. However, over 90% of serotonin is found outside the central nervous system (CNS) [54], and 5-HT_2A_ is expressed in several non-neuronal tissues, including vascular smooth muscle cells [55], the myometrial smooth muscle cells of pregnant human myometrium [45], and the uterine artery [56]. In addition, 5-HT_2A_ increases cell adhesion and cytoskeletal remodelling [57] and plays a significant role in platelet function [58]. Serotonin is also a key modulator of spinal nociceptive transmission [59] and may therefore have a role in endometriosis aetiology, pathophysiology, and pain perception, although the expression of these receptors in ectopic lesions has not been described.

Dopamine and its receptor (*DRD1–3*) are also neurotransmitters that have been associated with increased endometrial cancer severity and reduced progression-free survival [60], with the *DRD2* antagonist ONC201 shown to reduce tumour growth in vivo [61], including the highly aggressive *H3K27M* mutant gliomas [62]. *DRD2* is also expressed in human eutopic and ectopic endometrium [44], with *DRD2* polymorphisms identified as endometriosis candidate genes in women with peritoneal moderate/severe endometriosis [63]. Further to this, cabergoline, (a dopamine receptor agonist used for the treatment of hyperprolactinemia), was shown to reduce the size of endometriosis lesions via the inhibition of *VEGFR2* in a mouse model of endometriosis [64,65]. Cabergoline also induced a “lax stroma characteristic to atrophic or degenerative tissue”. In a more recent study, *DRD2* was upregulated in secretory phase tissue, with cabergoline inducing rapid stromal cell decidualization [66]. Taken together, the *DRD2* agonist cabergoline may terminally differentiate endometrial stromal cells via decidualization, leading to stromal cell death [67]. While a drug like cabergoline may have potential as an endometriosis therapeutic, it targets more than just *DRD2*, making a more selective *DRD2* inhibitor such as ONC201 more suitable for targeting *DRD2* in endometriosis. ONC201 was not part of this current screening library as it is not currently approved by the FDA and is listed as investigational.

This study was designed to demonstrate the suitability of using high-throughput compound screens to identify new therapeutics and targets for endometriosis. A limitation to this protocol was the use of eutopic endometrial stromal cells. As this protocol has generated significant robust data, future screens using ectopic lesion stromal cells can now be performed. Another limitation to this study is the use of stromal cells derived from the eutopic endometria of women diagnosed with endometriosis. To further improve and validate our findings and confirm that “normal” endometrial cells are not impacted by these compounds, the inclusion of eutopic endometrial stromal cells from healthy women with no pain, no endometriosis, and no other pathology should be included in future studies and screens.

## 5. Conclusions

This study has established the first quantitative high-throughput compound screening protocol for endometriosis and has identified several new compounds for translational consideration. While many of these compound hits and their molecular targets are well documented, there is some risk that the compounds will have off-target effects. The results of this study therefore establish a robust HTS process for evaluating novel molecules, peptides, and compounds as-endometriosis specific therapeutics. This study also demonstrates novel pathways and new molecular targets that may be significant contributors to endometriosis pathogenesis, inflammation, and nociception pain pathways and provides several new avenues of investigation for future endometriosis pathophysiological studies.

## Figures and Tables

**Figure 1 biomolecules-13-00965-f001:**
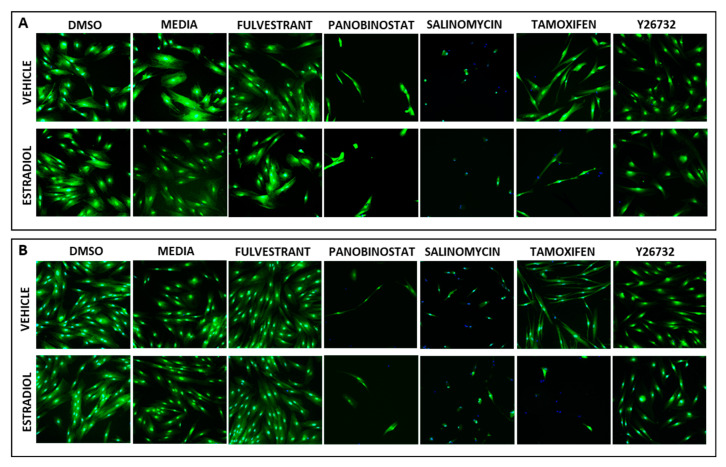
High-content imaging of QC wells. Images taken of 1458 (**A**) and 1455 (**B**) stromal cells exposed to DMSO, media, fulvestrant, panobinostat, salinomycin, tamoxifen, and Y26732. Cells are labelled with CFMDA (green) and DAPI (blue).

**Figure 2 biomolecules-13-00965-f002:**
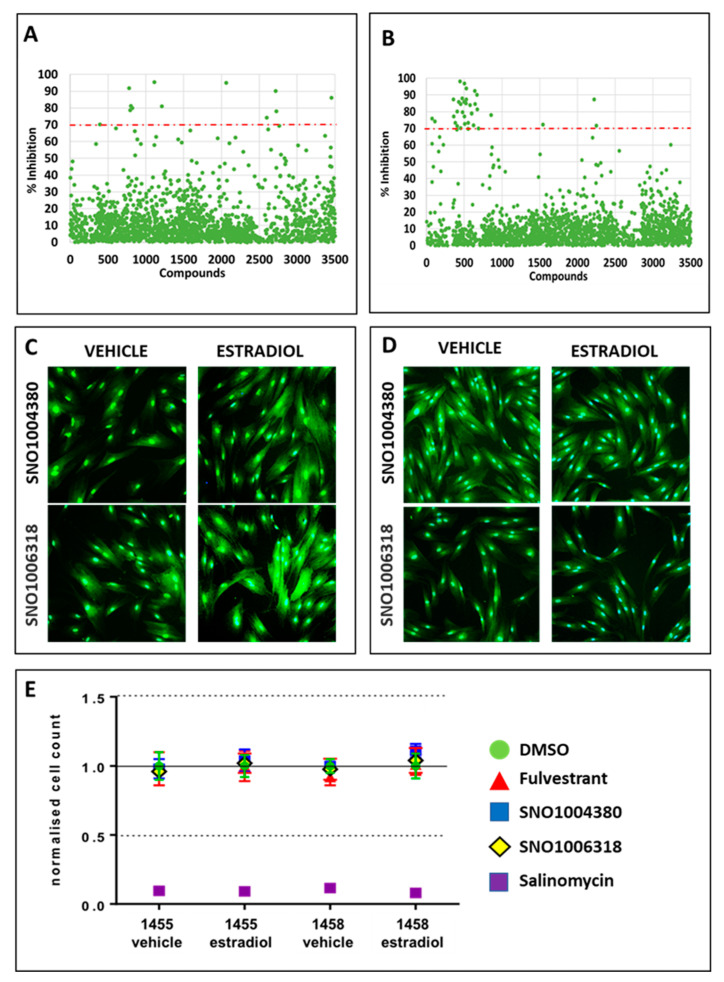
Initial qHTS, new QC compounds, and hits. The hESC lines 1455 (**A**) and 1458 (**B**) were screened against 3517 compounds in an HTS. Hits were identified as compounds that induced >70% cell growth inhibition (above red dashed line). New negative QC compounds SNO1004380 and SNO1006318 were selected from cell counts imaged from 1455 (**C**) and 1458 (**D**) to improve assay precision through the reliable and strong separation between positive and negative QC compound cell counts (**E**). Data are represented as means ± standard deviations.

**Figure 3 biomolecules-13-00965-f003:**
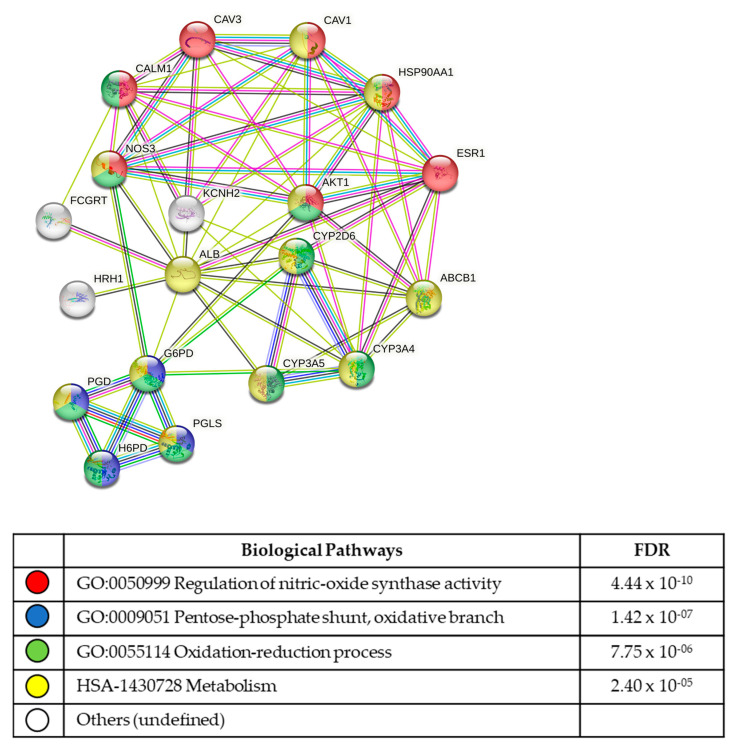
Molecular pathways identified for the hESC 1455 line. The 1455 cell line exhibited significant compound sensitivity to the regulation of nitric oxide synthase activity pathways, with an FDR ≤ 4.44 × 10^−10^.

**Figure 4 biomolecules-13-00965-f004:**
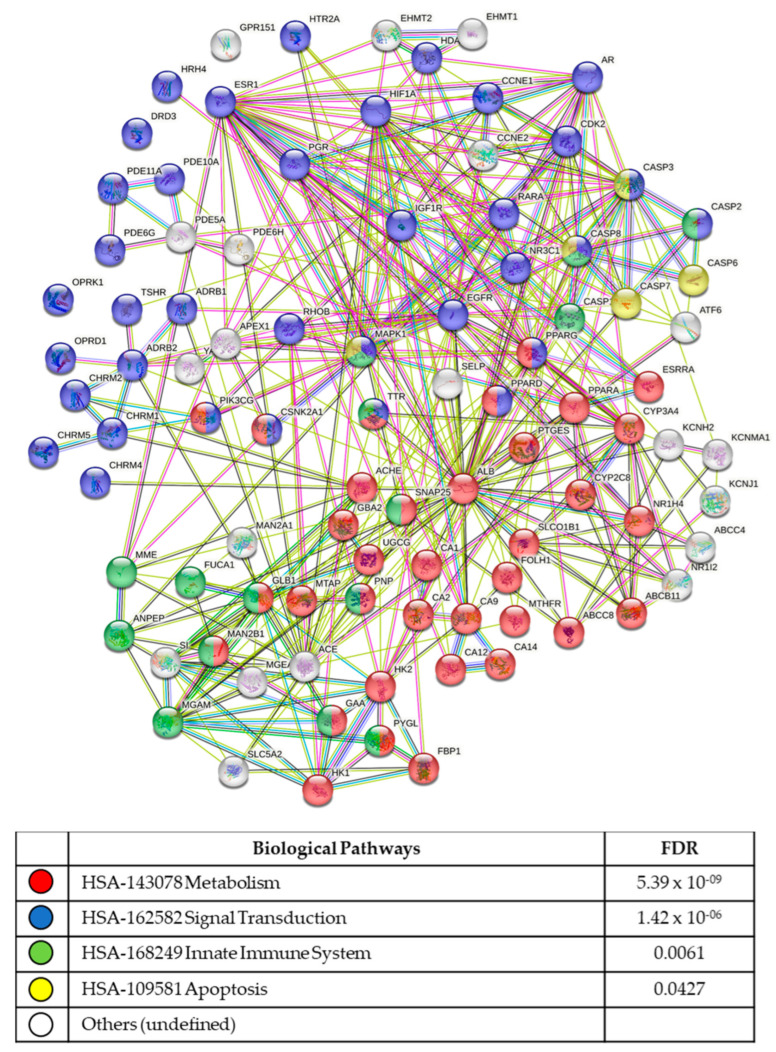
Molecular pathways identified for hESC 1458. The 1458 cell line exhibited significant compound sensitivity to metabolism pathways, with an FDR ≤ 0.5.39 × 10^−09^.

**Figure 5 biomolecules-13-00965-f005:**
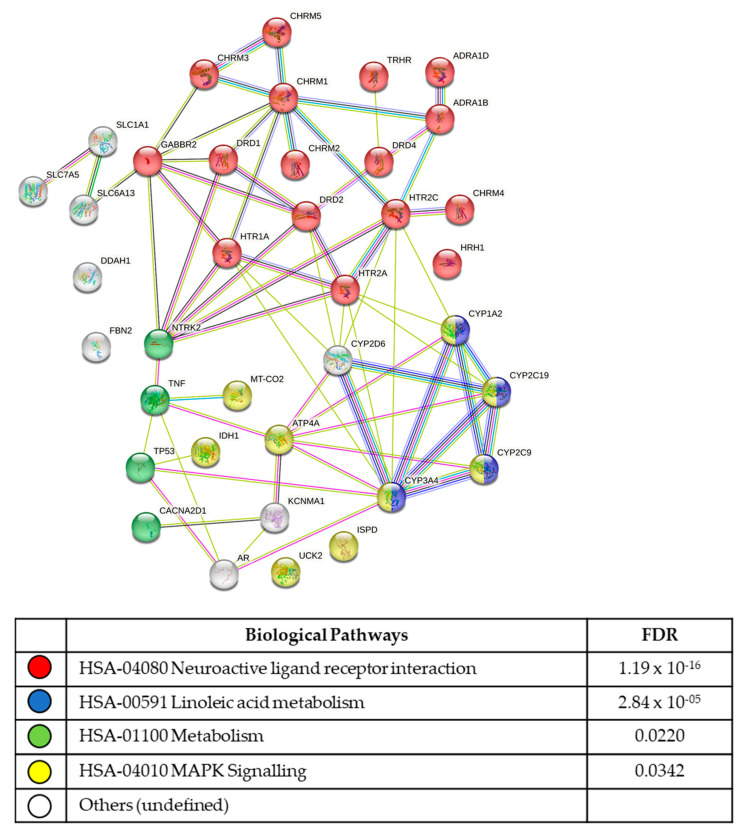
Molecular pathways common to both cell lines. The 1455 and 1458 hESC cells exhibited significant compound sensitivity to neuroactive ligand–receptor pathways, with a false discovery rate (FDR) ≤ 1.19 × 10^−16^.

**Table 1 biomolecules-13-00965-t001:** Lead compounds identified with IC_50_ from qHTS.

Compound Name	Generic Name	Compound Group	IC_50_ [µM] 1455	IC_50_ [µM] 1458
**SN01006330**	7,8-Dimethoxyflavone	anti-inflammatory	0.08	0.19
**SN01005561**	Aminothiazole	anti-microbial	0.12	2.08
**SN01005320**	Benazepril HCl	anti-hypertensive	9.37	0.06
**SN01005071**	Ceftibuten	anti-microbial	2.29	0.07
**SN00852779**	Chlordiazepoxide	anti-depressant	0.02	0.08
**SN01005451**	Chloroquinalol	anti-microbial	>20	0.08
**SN01006117**	Cytidine triphosphate disodium	anti-inflammatory	0.02	0.08
**SN01005419**	Editol	anti-inflammatory	ND	0.87
**SN01004366**	Hydroxyzine Pamoate	anti-histamine	0.02	ND
**SN01004587**	Indoprofen	anti-inflammatory	0.02	0.001
**SN01004583**	Ketotifen Fumarate	anti-histamine	0.22	ND
**SN01005061**	Pantoprazole	proton pump inhibitor	0.19	0.37
**SN01005391**	Pregabalin	anti-inflammatory	0.08	0.08
**SN01004486**	Promazine HCl	anti-psychotic	0.02	0.15
**SN01005316**	Repaglinide	anti-diabetic	>20	0.07
**SN01005445**	Sildenafil Citrate	anti-inflammatory	1.88	0.96
**SN01004511**	Spectinomycin HCl	anti-microbial	0.02	ND

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
