# Peer review of "Using a Quantitative High-Throughput Screening Platform to Identify Molecular Targets and Compounds as Repurposing Candidates for Endometriosis"

_biomolecules, 2023, doi:10.3390/biom13060965_

Round 1

Reviewer 1 Report

the main objective of this study was to identify possible molecular targets in patients with endometriosis. It is a novel approach which could help to identify some novel therapeutic targets in the future as many patients with endometriosis are treated empirically with non-steroid anti-inflammatory drugy. I have no comments regarding the methodology. The tables and figures are of appropriate quality. I have no additional comments.

Author Response

We thank Reviewer 1 for their comments.

Reviewer 2 Report

Churchill et al. presented some interesting data to identify molecular targets from endometrial stromal cell lines for repurposing of endometriosis. The new approach seems exciting, however, I have some major problems with the data.

Major Points

1.       In the Introduction (lines 47-63) there is a strong focus on endometriosis as an estrogen-dependent disease. Although this is more and more becoming the prevailing hypothesis, I just want to provide some food for thought: 1. As shown by Flores et al. 2018 (JCEM 103:4561) non-responders to progestin therapy showed a defective progesterone signaling. Thus, the PGR counteraction to estrogens seems to be impaired (progesterone resistance hypothesis); 2. Burns et al. 2018 (Endocrinology 159:103) found in an animal model that E2/ERα played a minor role in early lesion development; 3. Except for ovarian endometriosis the local E2 and E1 concentration in the tissues were not higher compared to endometrium; indeed they were even lower in peritoneal and deep infiltrating endometriosis (Huhtinen et al. 2012; JCEM 97:4228).

2.       Page 7, line 235. According to my knowledge salinomycin can increase the permeability of membranes for potassium ions and is apoptosis-inducing for cancer cells etc. I would not say that the mechanism is unknown.

3.       Fig. 2. How often did you repeat the experiments?

4.       Figs. 4 and 5. Because these data are very important I would suggest to use a better resolution and/or magnification. If this is not possible, please provide the data in a supplemental Excel file.

5.       The main reservations I have with this manuscript are the data in Fig. 5. I have checked all proteins/genes for the neuroactive ligand receptor interaction in the www.proteinatlas.org and found in endometrial glands and/or luminal cells: GABBR2, CHRM3, and DRD2. Only CHRM3 was localized also in stromal cells and/or vessels. All the other proteins are not found in the endometrium or only mRNA data are given in the proteinatlas database. Thus, for me it remains unclear why nearly all proteins you identified common to two stromal cell lines seem to be not expressed in the endometrial stromal cells. I am well aware that the proteinatlas database is not error-free or perfect, however, it gives us some hints. Furthermore, in the Discussion section, only DRD2 was found to be localized in the eutopic and ectopic endometrium, although in ref. 38 it remained unclear whether in the stromal and/or epithelial cells. Furthermore, they found a strongly decreased DRD2 expression in endometriosis, thus DRD2 as a therapeutic endometriosis target remains unclear to me due to the very low expression. As shown in ref 59 dopamine agonists modulated angiogenesis but not the stromal cells. In summary, your analysis of stromal cells to identify new compounds seem not to result in the clear identification of new stromal targets for endometriosis.

 Minor Points

1.       Page 2 line 65 – Please correct the typo and to an.

2.       Page 3 lines 112-121. If I would like to have the same compound library which number or identification (ID-No) should I use?

3.       Fig. 2C-E – Please replace the typo estrodiol by estradiol

Author Response

  1. In the Introduction (lines 47-63) there is a strong focus on endometriosis as an estrogen-dependent disease. Although this is more and more becoming the prevailing hypothesis, I just want to provide some food for thought: 1. As shown by Flores et al. 2018 (JCEM 103:4561) non-responders to progestin therapy showed a defective progesterone signaling. Thus, the PGR counteraction to estrogens seems to be impaired (progesterone resistance hypothesis); 2. Burns et al. 2018 (Endocrinology 159:103) found in an animal model that E2/ERαplayed a minor role in early lesion development; 3. Except for ovarian endometriosis the local E2 and E1 concentration in the tissues were not higher compared to endometrium; indeed they were even lower in peritoneal and deep infiltrating endometriosis (Huhtinen et al. 2012; JCEM 97:4228).

We thank Reviewer 2 for these comments. In response,

        Endometriosis is considered an estrogen dependent disease due to increased lesion growth in response to estrogen and a reduction in lesion growth due to reduced estrogen during pregnancy, after menopause or due to pharmacological suppression of estrogen secretion.  The article by Burns (2018) highlights this hormone dependence through the demonstration that following the Initiation Phase of lesion formation (< 3 days and immune cell/cytokine dependence), lesions become hormone dependent (Progression Phase) where E2 increased lesion weight and increased lesion numbers and weight in the absence of IL-6 due to an E2/IL6 cross talk.  Further, the Huhtinen (2012) article states estrogen-dependent growth of endometriosis lesions may be promoted by both systemic and locally synthesized E2 and that E2 concentrations in both serum and tissue is menstrual cycle dependent. They also described significant increases in E2 synthesis enzymes and ESR2 expression in ovarian and deep infiltrating endometriosis lesions suggesting an aberrant (disease associated) estrogen signaling pathway.  The Flores (2018) article is about predicting patient response to progestin treatment based upon progesterone receptor (PR) expression and found a lowered expression of PR in non-responders (Most likely due to PR promoter hyper-methylation (Wu et al., 2006)). Together, the papers presented by the reviewer, support our initial statement that endometriosis is an estrogen dependent disease and Lines 47-63 describing the estrogen signaling pathway is appropriate here.   

  1. Page 7, line 235. According to my knowledge salinomycin can increase the permeability of membranes for potassium ions and is apoptosis-inducing for cancer cells etc. I would not say that the mechanism is unknown.

We have updated the statement as follows “salinomycin is an antibiotic that interferes with membrane ion concentrations resulting in osmotic pressure imbalances and cell death (Mitani et al., 1976).

  1. 2. How often did you repeat the experiments?

Figure 2 is a representation of the initial screen that was performed once using a single concentration of drug to generate a list of compounds for the 5 point dilution screens for IC50 generation and these screens had 3 repeats.

  1. Fig 4 and 5. Because these data are very important I would suggest to use a better resolution and/or magnification. If this is not possible, please provide the data in a supplemental Excel file.

Supplementary Table III has been created with the list of targets identified.

  1. The main reservations I have with this manuscript are the data in Fig. 5. I have checked all proteins/genes for the neuroactive ligand receptor interaction in the www.proteinatlas.org and found in endometrial glands and/or luminal cells: GABBR2, CHRM3, and DRD2. Only CHRM3 was localized also in stromal cells and/or vessels. All the other proteins are not found in the endometrium or only mRNA data are given in the protein atlas database. Thus, for me it remains unclear why nearly all proteins you identified common to two stromal cell lines seem to be not expressed in the endometrial stromal cells. I am well aware that the protein atlas database is not error-free or perfect, however, it gives us some hints.

In response to the reviewers’ comments, we have added the following statement to the discussion

The qHTS was performed under estrogen conditions to mimic the diseased state of endometriosis.  In doing so, changes in gene expression induced by estrogen (doi: 10.3390/ijms16035864) result in compound targets that may be selectively expressed in ectopic lesions where estrogen signaling is aberrantly upregulated (doi: 10.1210/jc.2012-1154).

Furthermore, in the Discussion section, only DRD2 was found to be localized in the eutopic and ectopic endometrium, although in ref. 38 it remained unclear whether in the stromal and/or epithelial cells. Furthermore, they found a strongly decreased DRD2 expression in endometriosis, thus DRD2 as a therapeutic endometriosis target remains unclear to me due to the very low expression.  As shown in ref 59 dopamine agonists modulated angiogenesis but not the stromal cells. In summary, your analysis of stromal cells to identify new compounds seem not to result in the clear identification of new stromal targets for endometriosis.

In the discussion, ref 38 identified DRD2 mRNA in whole tissue samples and was not cell type specific.  Our statement was made to clarify that the target is expressed in this tissue.   While the expression of DRD2 in ectopic lesions was significantly lower compared to healthy endometrium, the increased expression in black lesions would still make DRD2 a valid target.  In regards to Ref 59, the following statement has been added to the text;

Cabergoline also induced a ‘lax stroma characteristic to atrophic or degenerative tissue’.  In a more recent study, DRD2 was found to be upregulated in secretory phase tissue with cabergoline inducing rapid stromal cell decidualisation (doi: 10.1210/clinem/dgab511 2021).  Taken together, the DRD2 agonist cabergoline may terminally differentiate endometrial stromal cells via decidualisation leading to stromal cell death (10.1530/REP-08-0539), making DRD2 a promising endometriosis target”.   

Minor Points

  1. Page 2 line 65 – Please correct the typo and to an.

Correction made

  1. Page 3 lines 112-121. If I would like to have the same compound which number or identification (ID-No) should I used library?

The following statement has been added to the text; These plates were the CA-FDA Collection, the CA-Epigenetics Collection, and the CA-Kinase Collection.

  1. 2C-E – Please replace the typo estrodiol by estradiol

The figure has been corrected

Reviewer 3 Report

Globally, the work is devoted to the search for new drugs for the treatment of endometriosis. As noted in the Summary, “This study demonstrates for the first time the feasibility of performing large compound screens for the identification of new translatable therapeutics and improved characterization of endometriosis molecular pathophysiology”. Using the HTS platform the authors were able to isolate several drugs that inhibit the growth of cultures of immortalized stromal cells of the eutopic endometrium of two patients with endometriosis in the presence of estradiol.

The design of the experiment limits the practical value of this work. First of all, the adequacy of using cultures of immortalized stromal cells of the eutopic endometrium as a therapeuticus taget raises questions. In the treatment of endometriosis , the isdeal goal is to suppress the growth of the ectopic endometrium while maintaining  the function of the eutopic, and many studies indicate that these tissues, while similar in appearance, are quite differente at the molecular level.

Secondly, in order to improve the “characterization of endometriosis molecular pathophysiology”, a control should be included in the study – cultures of immortalized stromal cells of the eutopic endometrium of healthy women. These limitations should be discussed in the text. After making such changes the work may be recommended for publication.

Two small remarks. The statement that «Endometriosis …affects up to 11.4% of women of reproductive age and gender diverse people with a uterus» (line 15) is correct in form, but not entirely true in substance, because the presence of the uterus is not necessary for the development of endometriosis – it happens in persons with Rokitansky syndrome, in patients after histerectomy and even occasionally in men. In addition, EstrOdiol is mentioned twice in Fig.1

Author Response

The design of the experiment limits the practical value of this work.  First of all, the adequacy of using cultures of immortalized stromal cells of the eutopic endometrium as a therapeuticus taget raises questions. In the treatment of endometriosis, the isdeal goal is to suppress the growth of the ectopic endometrium while maintaining  the function of the eutopic, and many studies indicate that these tissues, while similar in appearance, are quite differente at the molecular level.

In response to these comments, we have included the following statement in the Discussion;

“This study was designed to demonstrate the suitability of using high throughput compound screens to identify new therapeutics and targets for endometriosis. A limitation to this protocol was the use of eutopic endometrial stromal cells.   As this protocol has generated significant robust data, future screens using ectopic lesion stromal cells can now be performed.”

Secondly, in order to improve the “characterization of endometriosis molecular pathophysiology”, a control should be included in the study – cultures of immortalized stromal cells of the eutopic endometrium of healthy women. These limitations should be discussed in the text. After making such changes the work may be recommended for publication.

In response to these comments, we have included the following statement in the Discussion; 

“Another limitation to this study is the use of stromal cells derived from the eutopic endometrium of women diagnosed with endometriosis.  To further improve and validate our findings and confirm that ‘normal’ endometrial cells are not impacted by these compounds, the inclusion of eutopic endometrial stromal cells from healthy women with no pain, no endometriosis and no other pathology should be included in future studies and screens.”   

Two small remarks. The statement that «Endometriosis …affects up to 11.4% of women of reproductive age and gender diverse people with a uterus» (line 15) is correct in form, but not entirely true in substance, because the presence of the uterus is not necessary for the development of endometriosis – it happens in persons with Rokitansky syndrome, in patients after histerectomy and even occasionally in men.

In response to the reviewers comments, the following statement has been added to the introduction;

“In extremely rare occasions, endometriosis can occur in women with Rokitansky-Küster-Hauser syndrome (MRKH) syndrome (with an absent or rudimentary non-functioning uterus)  (DOI: 10.1530/REP-19-0106), and in men (doi: 10.1155/2018/2083121).”

In addition, EstrOdiol is mentioned twice in Fig.1

The figure has been corrected

Round 2

Reviewer 2 Report

Thanks for all your answers to my queries. I have only two minor remarks to the Introduction, lines 35-37.

In Ref 3 it is clearly stated that only with a rudimentary uterus endometriosis was observed. Until today no one has ever proved that without a uterus endometriosis can occur.

In Ref 4 the staining of the "endometrial-like" tissue with CD10 is unconvincing, also the other markers used are not highly specific endometrial markers. Furthermore, the occurrence of endometrial-like tissue in men is very often not proved by the use of highly specific endometrial markers. In my opinion if the occurrence of endometrial tissue in males can be proven unequivocally even then it is an exception to the Sampson hypothesis but not a counter argument.

Author Response

We agree with the comments related to Ref3 and have made the following amendments to the introduction

In extremely rare conditions such as Rokitansky-Küster-Hauser (MRKH) syndrome (congenital aplasia of the vagina, cervix and uterus), endometriomas have been identified most likely due to retrograde menses due to remnants of endometrial tissue [3].   Further to this, endometriosis of the bladder, peritoneum and genitals has been described in men in men [4].  While more substantial evidence of endometriotic lesions in males is needed, these rare cases may be demonstrations of remnant embryological cell growth [4] or mesothelium metaplasia [5] in response to elevated estrogen.  

Please see attachement for amended manuscript
